# Exosomal and Soluble Programed Death-Ligand 1 (PD-L1) Predicts Responses to Pembrolizumab in Patients with Extranodal NK/T-Cell Lymphoma

**DOI:** 10.3390/cancers14225618

**Published:** 2022-11-16

**Authors:** Seok Jin Kim, Kyung Ju Ryu, Bon Park, Sang Eun Yoon, Junhun Cho, Yoon Park, Won Seog Kim

**Affiliations:** 1Division of Hematology-Oncology, Department of Medicine, Samsung Medical Center, Sungkyunkwan University School of Medicine, Seoul 06351, Republic of Korea; 2Department of Health Sciences and Technology, Samsung Advanced Institute of Health Science and Technology, Sungkyunkwan University, Seoul 06351, Republic of Korea; 3Department of Pathology, Samsung Medical Center, Sungkyunkwan University School of Medicine, Seoul 06351, Republic of Korea; 4Biomedical Research Institute, Korea Institute of Science and Technology, Seoul 02792, Republic of Korea

**Keywords:** soluble PD-L1, exosome, extranodal NK/T-cell lymphoma, pembrolizumab

## Abstract

**Simple Summary:**

Soluble and exosomal programed death-ligand 1 can be upregulated in extranodal natural killer/T-cell lymphoma. We investigated the association between pre-treatment soluble and exosomal programed death-ligand 1 and outcomes in extranodal natural killer/T-cell lymphoma patients who received pembrolizumab as a salvage treatment. Treatment outcomes and overall survival after pembrolizumab were significantly better in patients with low soluble and exosomal programed death-ligand 1. Thus, soluble and exosomal programed death-ligand 1 can predict responses to pembrolizumab, making it a useful pre-treatment biomarker for extranodal natural killer/T-cell lymphoma patients receiving pembrolizumab.

**Abstract:**

Soluble and exosomal programed death-ligand 1 (PD-L1) can be upregulated in extranodal natural killer/T-cell lymphoma (ENKTL). However, its clinical role in predicting outcomes after pembrolizumab treatment has yet to be studied in ENKTL patients. We investigated the association between pre-treatment soluble and exosomal PD-L1 and outcomes in ENKTL patients who received pembrolizumab as a salvage treatment. The production of soluble and exosomal PD-L1 was analyzed in vitro using an etoposide-resistant ENKTL cell line. Serum levels of soluble and exosomal PD-L1 were measured in patients with relapsed or refractory ENKTL prior to treatment with pembrolizumab. Relapsed or refractory ENKTL patients who received pembrolizumab as a salvage therapy between May 2017 and March 2021 were analyzed at our institute. Soluble and exosomal PD-L1 was significantly higher in serum samples of relapsed or refractory ENKTL patients compared with healthy controls, which is consistent with increased production of soluble and exosomal PD-L1 in an etoposide-resistant ENKTL cell line (SNK6R), which was found to show increased expression of soluble and exosomal PD-L1. Serum-soluble PD-L1 levels were significantly correlated with exosomal PD-L1, and were significantly lower in responders to pembrolizumab compared with non-responders. Longitudinal analysis after pembrolizumab also revealed a relationship between PD-L1 levels and responses. Treatment outcomes and overall survival after pembrolizumab were significantly better in patients with low soluble and exosomal PD-L1. In conclusion, soluble and exosomal PD-L1 can predict responses to pembrolizumab in ENKTL patients, making it a useful pre-treatment biomarker for ENKTL patients receiving pembrolizumab.

## 1. Introduction

The identification of cell-targeted immune checkpoint inhibition is a revolutionary milestone in oncology [1,2]. Programed death-ligand 1 (PD-L1) is a primary ligand of programed cell-death protein 1 (PD1), and the interactions between PD-L1 on tumor cells and PD1 on effector T cells can induce T-cell exhaustion, leading to evasion of a host’s immune response against tumor cells [3]. PD1 inhibitors such as pembrolizumab and nivolumab can inhibit this interplay between PD1 and PD-L1, resulting in anti-tumor effects in various solid cancers [4,5]. However, the efficacy of PD1 inhibitors is unsatisfactory in the majority of patients with non-Hodgkin lymphoma (NHL) [6]. As a result, PD1 inhibitors are used primarily in a few NHL subtypes expressing PD-L1. 

Extranodal natural killer (NK)/T-cell lymphoma (ENKTL) responds to pembrolizumab because the integration of the Epstein–Barr virus (EBV) in tumor cells can induce PD-L1 expression [7,8]. However, not all patients with ENKTL respond, and primary refractoriness to immune checkpoint inhibitors has also been reported [8]. Although analysis of tumor tissue for PD-L1 expression and tumor mutation volume have been used to predict responses to immune checkpoint inhibitors, a tumor tissue biopsy may not always be possible for such analyses. In addition, intra-patient heterozygosity according to the involved sites and time of obtaining tumor tissue can influence the predictive value of archived tissue samples. A liquid biopsy via peripheral blood sampling may, therefore, be a more efficient and convenient method of developing biomarkers for predicting responses to immune checkpoint inhibitors in ENKTL patients. 

Soluble PD-L1 in blood has been studied as a biomarker for liquid biopsies, along with histological PD-L1 expression in tumor tissue, and high soluble PD-L1 reportedly relates to poor prognoses in various cancers, including lymphomas [9,10,11]. Soluble PD-L1 is thought to be derived from PD-L1–positive cells, including tumor cells, and exosomes released from PD-L1–positive cells can be a source of soluble PD-L1 [12,13]. Release of exosomes carrying PD-L1 on their surface has been demonstrated in metastatic melanoma, and exosomal PD-L1 may be able to suppress CD8 T cells that facilitate tumor growth [14]. In addition, patients with melanoma who failed to respond to PD1 inhibitors were reported to have elevated levels of exosomal PD-L1 [14]. In ENKTL, PD-L1 expression can be upregulated by EBV-driven latent membrane protein 1 (LMP1) through the nuclear factor kappa B pathway and is associated with poor prognoses [15]. High levels of soluble and exosomal PD-L1 have been reported in ENKTL patients with poor prognoses [16,17]. Thus, a previous retrospective study reported soluble PD-L1 as a predictor of chemotherapy response in ENKTL patients [18]. However, the relationship with the response to pembrolizumab has yet to be studied in ENKTL patients. Here, we investigate the association between pre-treatment soluble and exosomal PD-L1 and outcomes in ENKTL patients who received pembrolizumab as a salvage treatment.

## 2. Materials and Methods

### 2.1. Study Design

Most treatment regimens for newly diagnosed ENKTL patients contain etoposide because of its activity against EBV-positive lymphoma cells. Relapses or refractory ENKTL in patients previously exposed to etoposide implies the presence of etoposide-resistant lymphoma cells. As immune checkpoint inhibitors are primarily used as salvage therapy for patients who relapse after etoposide-containing chemotherapy, we first analyzed expression of PD-L1 in an etoposide-resistant ENKTL cell line.

An SNK6 cell line resistant to etoposide (SNK6R) was developed at concentrations of 200 nM and 400 nM, as we previously reported [19]. The production of soluble PD-L1 and exosomal PD-L1 was measured in a culture media of SNK6R, and the transfer of PD-L1 via exosomes derived from SNK6R to macrophages was analyzed by co-culturing SNK6R with a human monocyte cell line (THP-1). We then measured the amount of soluble PD-L1 and exosomal PD-L1 in archived serum samples obtained from relapsed or refractory ENKTL patients before initiating pembrolizumab treatment. The treatment outcome of patients after pembrolizumab was analyzed according to the pre-treatment concentration of soluble PD-L1 and exosomal PD-L1. Post-treatment serial changes were also analyzed. 

### 2.2. Patients

Relapsed or refractory ENKTL patients who received pembrolizumab as a salvage therapy between May 2017 and March 2021 were analyzed at our institute, and were registered into our prospective cohort study after supplying written informed consent (IRB No. 2016-11-040, NCT03117036). Doses of 100 or 200 mg of pembrolizumab were administered every three weeks, and the response was assessed using 2-deoxy-2-fluorine-18-fluoro-D-glucose (^18F^-FDG) positron emission tomography integrated with computed tomography (PET/CT) and CT scans based on recommendations for initial evaluation, staging, and response assessment of Hodgkin and NHL using the Lugano classification [20]. If a complete response (CR), partial response (PR), or stable disease (SD) was documented, treatment was continued, and the response evaluation was repeated every three cycles until disease progression was confirmed. The pre-treatment serum samples were archived before administration of pembrolizumab, and post-treatment serum samples were collected at the time of interim and final-response evaluations after the completion of the second cycle, and then stored at −80 °C. Prior to administering pembrolizumab, pre-treatment disease status was evaluated with PET/CT and/or CT scans and blood tests, including serum lactate dehydrogenase and a blood EBV DNA titer. Based on five variables (age > 60 years, stage III/IV disease, distant lymph-node involvement, non-nasal type, and blood EBV DNA), the pre-treatment prognostic index for NK lymphoma (PINK-E) was determined as low (1 or no factor), intermediate (two factors), or high-risk groups (≥three factors) [21]. Pre-treatment disease status was dichotomized into relapsed disease, which was designated as relapsed disease occurring during follow-up, or refractory disease, which was designated as disease refractory to the last treatment prior to enrollment. 

### 2.3. Cell Culture 

SNK6 and SNK6R cells were cultured in RPMI-1640 medium supplemented with heat-inactivated 10% fetal bovine serum (FBS), recombinant human interleukin-2 (PeproTech, Rocky Hill, NJ, USA), and treated with etoposide (Sigma-Aldrich, St Louis, MO, USA) to maintain drug resistance. For trans-well co-culture assays, THP-1 cells (ATCC, Rockville, MD, USA) were seeded into a 24-well plate (lower chamber, 3 × 10^5^) and differentiated into macrophages (M0) with 150 nM phorbol 12-myristate 13-acetate (PMA, Sigma-Aldrich) for 24 h, after which the PMA-containing medium was replaced with 10% FBS containing a 1 × RPMI medium for 24 h. The SNK6R cells were then plated in a trans-well chamber (pore size, 0.4 μm; Corning Costar, Tewksbury, MA, USA) and incubated for 48 h. Cell densities were chosen to ensure a 1:1 ratio between SNK6R and THP-1 cells co-cultured in a 10% exosome-depleted FBS-cultured RPMI medium. Macrophage M2 polarization was obtained by incubation with 20 ng/mL of interleukin-4 (R&D Systems, Minneapolis, MN, USA) and 20 ng/mL of interleukin 13 (R&D Systems, USA) as previously reported [22].

### 2.4. RNA Stability Assay and Real-Time Quantitative Polymerase Chain Reaction 

To determine the stability of PD-L1, cells were treated with actinomycin D (1 μg/mL) and harvested at the indicated times. Total RNA was isolated using the TRIzol reagent (Invitrogen, Carlsbad, CA, USA) according to the manufacturer’s instructions, and 1 μg of RNA was converted to cDNA using an Omniscript RT kit (Qiagen, Valencia, CA, USA). Quantitative real-time polymerase chain reaction amplification was performed using TaqMan gene expression assays, a TaqMan expression master mix, and QuantStudio6 (Applied Biosystems, Foster, CA, USA). The TaqMan gene expression assays included PD-L1 (Hs00204257_m1), GAPDH (Hs02786624_g1), and 18S rRNA (Hs03003631_g1) (Applied Biosystems). Relative expression of mRNA was calculated by the 2^−ΔΔCt^ method, and 18S rRNA was used for normalization.

### 2.5. Western Blot Analysis

Cells were lysed in a RIPA buffer (0.5% sodium deoxycholate, 1% Nonidet P-40, 150 mM NaCl, 50 mM Tris [pH 7.5] and 0.1% sodium dodecyl sulfate [SDS]) containing complete protease and phosphatase inhibitors. In total, 30 μg of protein samples was electrophoresed on a 4–12% SDS polyacrylamide gel and transferred to nitrocellulose membranes. After introducing a blocking solution of 5% non-fat milk for 1 h at room temperature, membranes were incubated with primary antibodies including anti-pAKT (1:1000), anti-AKT (1:3000), anti-pERK (1:1000), anti-ERK (1:3000), p53 (1:1000), PD-L1 (1:1000) (Cell Signaling, Danvers, MA, USA) and β-actin (1:5000) (Sigma-Aldrich, St Louis, MO, USA) overnight at 4 °C. The blots were washed with TBST buffer (50 mM Tris [pH 7.5], 150 mM NaCl. 0.05% Tween 20) and incubated with secondary antibodies (1:5000) for 1 h. Proteins were visualized using an enhanced chemiluminescence reagent (Invitrogen, Carlsbad, CA, USA).

### 2.6. Isolation and Characterization of Exosome 

To isolate exosomes from the cell culture medium, cells were suspended in RPMI-1640 medium containing 10% exosome-depleted FBS for 72 h. Then, cell culture medium was collected and differentially centrifuged at 300× *g* at 4 °C for 10 min, 2000× *g* at 4 °C for 10 min, 10,000× *g* at 4 °C for 30 min, and ultracentrifuged at 100,000× *g* at 4 °C for 120 min. Then, the pellet was washed in PBS and was secondly ultracentrifuged at 110,000× *g* at 4 °C for 70 min. After the supernatant was removed, the final pellet was resuspended in PBS. Serum samples with volumes of 100 μL were thawed on ice and differentially centrifuged at 2000× *g* at 4 °C for 10 min, 10,000× *g* at 4 °C for 30 min, and passed through a 0.22-μm filter to remove cell debris. Clarified serum mix with ExoQuick (System Biosciences, Palo Alto, CA, USA) was incubated at 4 °C for 30 min then centrifuged twice at 1500× *g* for 30 and 5 min, respectively, to remove the supernatant. The pellets were then resuspended in 100 μL of PBS. The ExoView Tetraspanin chip (ExoView, Boston, MA, USA) was arrayed with antibodies against proteins CD81, CD63, and CD9 markers of exosomes, and HLA immunoglobin G1 was used as a negative control. The samples were dropped onto the chip surface and placed at room temperature overnight. The cellular origin of the exosomes was analyzed using ExoView Tetraspanin Labeling ABs (EV-TC-AB-01) of anti-CD81/ALEXA 555, anti-CD63/ALEXA 647, anti-CD9/ALEXA 488, and anti-PD-L1/APC. Images of isolated exosomes were analyzed using an ExoView R100 reader and ExoScan 2.5.5 acquisition software (ExoView, Boston, MA, USA). The data were analyzed using ExoViewer 2.5.0, with sizing thresholds set to a diameter of 50 to 200 nm. For nanoparticle tracking analysis (NTA), exosomes were examined using a Nanosight NS300 (NanoSight Ltd., Amesbury, UK), and the size and quantity of particles were analyzed using NTA software (version 2.3; NanoSight Ltd.). For transmission electron microscopy (TEM), exosomes were fixed in 2% paraformaldehyde and transferred onto Formvar-carbon-coated EM grids. Fixed samples were allowed to absorb for 10 min in a dry environment, and grids were rinsed in PBS. Grids were washed 10 times with distilled water, negative stained with 1% uranyl acetate for 1 min, and then air-dried. Grids were observed using a Hitachi 7700 transmission electron microscope operated at 80 kV.

### 2.7. Measurement of Soluble PD-L1 and Exosomal PD-L1 

Soluble PD-L1 and exosomal PD-L1 levels were measured by an enzyme-linked immunosorbent assay (ELISA), using a PD-L1/B7-H1 Quantikine ELISA kit (cat. no. DB7H10, R&D systems, Minneapolis, MN, USA). In brief, 100 μL of standards at different concentrations and samples were added to the wells and incubated for 2 h at room temperature. The wells were washed four times, after which B7-H1 conjugate was added to each well, which was incubated for 2 h at room temperature and then washed again. Next, substrate solution was added and incubated for 30 min in the dark. Stop solution was added and the absorbance (optical density [OD]) of each well was measured using a Spectramax microplate reader with a wavelength of 450 nm (ABS Plus, San Jose, CA, USA). The ELISA detection limit was 4.52 pg/mL. Standards and all the samples were tested in duplicate.

### 2.8. Flow Cytometry 

Patients’ peripheral blood mononuclear cells were stained with 5 uL of fluorochrome-conjugated anti-CD3 (HIT3a), anti-CD4 (OKT4), and anti-CD8 (SK1) for 30 min at 4 °C after blocking with anti-human TruStain FcX (BioLegend) in a fluorescent activated cell-sorting buffer (PBS containing 1% BSA and 0.1% sodium azide). Data acquisition used a CytoFLEX flow cytometer (Beckman Coulter, Brea, CA, USA) and analysis was carried out using FlowJo (v.10.5.3, TreeStar, Woodburn, OR, USA).

### 2.9. Immunohistochemistry 

All patients were pathologically diagnosed with ENKTL based on a positive result for EBV in situ hybridization (ISH) by a lymphoma pathologist (J.H.). Immunohistochemistry was performed for PD-L1 as described in Appendix A. Tumor cells expressing PD-L1 and PD-L1–positive tumor-associated macrophages (TAMs) were evaluated because the expression of PD-L1 in TAMs has been reported in ENKTL patients [17]. To represent the extent of PD-L1 expression where tumor cells were present, the percentage of total PD-L1–positive cells, including PD-L1–positive tumor cells and TAMs among tumor cells, was calculated as a PD-L1 score. A score > 10 was designated as high PD-L1 expression, based on our previous study of immune subtyping of ENKTL [23]. 

### 2.10. Statistical Analyses

Chi-square tests were used to analyze the association of laboratory parameters with treatment response, and Spearman bi-variate correlation analysis was conducted to analyze the association between soluble and exosomal PD-L1. The median potential follow-up time with 95% confidence intervals was determined using the Kaplan–Meier method [24]. The time to progression was defined as the first date of the first cycle of pembrolizumab treatment to the date of disease progression. Overall survival after pembrolizumab treatment was the time between the first date of pembrolizumab treatment and the date of any kind of death or last follow-up date. Survival was estimated based on Kaplan–Meier curves and compared using log-rank tests. Two-sided statistical tests yielding a *p* value < 0.05 were considered significant. Statistical analyses were performed using IBM PASW version 24.0 software (SPSS Inc., Chicago, IL, USA).

## 3. Results

### 3.1. Increase of Soluble and Exosomal PD-L1 in Etoposide-Resistant Cells 

The half-maximal inhibitory concentration (IC_50_) of SNK6R was higher than that of control SNK6 cells (Figure 1A). The expression of phosphorylated AKT and ERK as well as p53 and LMP1 was higher in SNK6R cells than control SNK6 (Figure 1B). The expression of PD-L1 was also increased in SNK6R cells, and enhanced expression of PD-L1 was evident on the surface of SNK6R cells (Figure 1B,C). In a culture media of SNK6R cells, the level of soluble PD-L1 was higher compared with that of control cells, particularly at a concentration of 400 nM (Appendix A). A comparison of mRNA stability between SNK6R and SNK6 cells revealed persistently increased expression of PD-L1 in the SNK6R cells at an etoposide concentration of 400 nM (Appendix A). We isolated exosomes from the culture media of SNK6R (Figure 1D), and sorted them using exosomal markers, CD63, CD81 and CD9, and SNK6R-derived exosomes showed higher expression of PD-L1 compared with SNK6 cells regardless of the type of exosomal markers (Appendix A). The co-culture of SNK6R with THP-1 cells showed higher expression of PD-L1 compared with control (M0 and M1 THP-1) cells, supporting the transfer of exosomal PD-L1 from SNK6R to THP-1 cells (Figure 1E).

### 3.2. Soluble and Exosomal PD-L1 in Relapsed or Refractory ENKTL Patients 

The median age of 30 patients was 51 years (range: 23–75), and more than half of patients had an Eastern Cooperative Oncology Group (ECOG) performance status of 2 or greater (Table 1). Out of 30 patients who received pembrolizumab as a salvage treatment for their relapsed or refractory ENKTL, five were previously treated with a PD-L1 inhibitor, avelumab, in our previous phase II study [25]. Those patients received pembrolizumab after they failed to respond to avelumab. Measurements of soluble PD-L1 in pre-treatment serum samples ranged between 42 and 2475 pg/mL, and patients could be dichotomized based on 100 pg/mL (Figure 2A). These levels of soluble PD-L1 were significantly higher than those of a healthy control group (median: 48 pg/mL, range: 39–58). Exosomes were also isolated from 1 mL serum samples that were archived prior to administration of pembrolizumab. The median concentration of exosomal PD-L1 was 10 pg/mL (range: 1.7–1250.8), which was higher than that of a healthy control group (median: 1.7 pg/mL, range: 1.4–5.9). Exosomal PD-L1 levels were significantly correlated with soluble PD-L1 levels (Figure 2B). Three patients who were previously treated with avelumab showed markedly elevated levels of soluble PD-L1 (>2300 pg/mL) and exosomal PD-L1 (Figure 2A,B). The median size and number of exosomes were 78 nm (range: 62–119 nm) and 38.1 × 10^12^/mL (range: 14.0–113.8 × 10^12^; Figure 2C). When patients were dichotomized into high and low exosomal PD-L1 according to median values, there was no significant association with PD-L1 scores in tumor tissue (*p* = 0.484, Figure 2D). Most patients with high levels of soluble PD-L1 (>100 pg/mL) had a high PD-L1 score, although this relationship was not statistically significant (*p* = 0.096; Figure 2E). In the bivariate analysis, blood EBV DNA titers were not associated with levels of soluble and exosomal PD-L1 (*p* > 0.05). Patients with an EBV DNA titer >10,000 IU did not exhibit a significant relationship with high soluble and exosomal PD-L1 (Figure 2F).

### 3.3. Response to Pembrolizumab and Levels of Soluble and Exosomal PD-L1

Based on the best response, four patients achieved a CR and seven patients showed a PR, whereas the remaining 19 patients developed PD during pembrolizumab treatment. Pre-treatment exosomal and soluble PD-L1 were associated with lower levels in patients with CR and PR compared with patients with PD (Figure 3A). When patients were dichotomized into low and high exosomal PD-L1 groups based on median values, most clinical and laboratory characteristics prior to pembrolizumab did not significantly differ between low and high exosomal PD-L1 groups (Table 1). However, patients with high exosomal PD-L1 had lower ECOG performance status scores and were more refractory to previous treatments. The response to pembrolizumab was therefore significantly related to pre-treatment low exosomal PD-L1 (Figure 3B). Likewise, patients with low levels of soluble PD-L1 showed a significant association with treatment response (Figure 3B). Out of five patients who were previously treated with avelumab, all but one belonged to the high soluble and exosomal PD-L1 group and failed to respond, whereas only one patient with low soluble and exosomal PD-L1 achieved PR. Patients achieving CR or PR could receive pembrolizumab up to >25 cycles, and most patients belonged to the low exosomal PD-L1 group (Figure 3C). The time to progression after pembrolizumab was significantly longer in the low exosomal PD-L1 group compared with the high exosomal PD-L1 group (Figure 3D). Patients with low soluble PD-L1 also had longer time to progression than the high soluble PD-L1 group (*p* = 0.017). Accordingly, overall survival after pembrolizumab was significantly better in patients with low soluble PD-L1 compared with high soluble PD-L1 (Figure 3E). Pre-treatment exosomal PD-L1 also showed a similar association with post-treatment overall survival, although it did not reach statistical significance (*p* = 0.068).

### 3.4. Longitudinal Analysis before and after Pembrolizumab Treatment

The post-treatment changes of soluble and exosomal PD-L1 were also evaluated in patients who had longitudinally archived serum samples. The first case had localized relapsed disease and low levels of soluble PD-L1 before pembrolizumab treatment. The patient achieved CR, with lower serum levels of soluble PD-L1 after treatment (Figure 4A). The second case with stage IV disease showed disease progression after the eighth cycle, with increased levels of soluble PD-L1 (Figure 4B). Flow cytometry using peripheral blood mononuclear cells showed an increased frequency of CD4 and CD8 T cells after pembrolizumab treatment in patients with low levels of exosomal and soluble PD-L1 achieving CR (Figure 5A). However, patients with high exosomal and soluble PD-L1 showed progression without an increase in CD4 and CD8 cells (Figure 5B). Patients experiencing CR had more CD4 and CD8 cells after treatment, whereas patients with PD had fewer CD4 and CD8 T cells after treatment (Figure 5C).

## 4. Discussion

In this study, we first established the etoposide-resistant cell line (SNK6R) to make an in vitro model mimicking clinically aggressive ENKTL and to evaluate its association with PD-L1 expression. SNK6R showed the increased expression of phosphorylated AKT and this result was consistent with a previous in vitro study reporting the phosphorylated AKT-mediated proliferation and viability of SNK6 cells [26]. The previous transcriptomic characterization of ENKTL also reported in the frequent mutations of JAK/STAT and p53 in its molecular subtype, which was named as TISM type where PD-L1 expression was increased [27]. Considering LMP1 was also related with PD-L1 expression, the etoposide-resistant SNK6R could show enhanced expression of PD-L1 on its surface and release soluble and exosomal PD-L1 (Figure 1B,C). In addition, our co-culture of SNK6 with THP-1 cells showed the expression of PD-L1 on THP-1 increased when THP-1 cells were co-cultured with SNK6R compared to the control group (Figure 1E). Although our co-culture with SNK6R and THP-1 cells could not exactly represent in vivo situation due to the presence of various factors influencing the expression of PD-L1 on macrophages, the transfer of exosomes carrying PD-L1 on their surface from tumor cells to macrophages might occur in our experiment considering the well-known role of exosomes in the crosstalk between tumor cells and tumor-associated macrophages [28]. These results are consistent with previous studies reporting an association of soluble and exosomal PD-L1 with poor treatment outcomes in ENKTL patients, and PD-L1 expression on both malignant cells and tumor-infiltrating macrophages [16,17].

Although the mechanism responsible for the production of soluble PD-L1 has yet to be determined, proteolytic cleavage by endogenous matrix metalloproteinase could generate soluble PD-L1 from membrane-bound PD-L1 [29,30]. In addition to soluble PD-L1, extracellular PD-L1 could exist in another important form—exosomal PD-L1—because tumor cells can produce and secrete exosomes with cancer-promoting contents [31]. Enhanced expression of membrane-bound PD-L1 in tumor cells, therefore, may result in an increase in circulating soluble PD-L1 and tumor cell–derived exosomes carrying PD-L1. Levels of soluble and exosomal PD-L1 in patients with relapsed or refractory ENKTL were significantly higher than in a healthy control group. In particular, three patients previously treated with avelumab showed significantly higher levels of soluble and exosomal PD-L1 compared with the other ENKTL patients (Figure 2A). These elevated levels may be related to previous exposure to avelumab, although few studies of post-treatment changes in circulating soluble and exosomal PD-L1 after treatment with a PD-L1 inhibitor have been published. When levels of soluble and exosomal PD-L1 were compared with the extent of PD-L1 expression in tumor tissues, patients with more than 100 pg/mL of soluble PD-L1 tended to have a high PD-L1 score, although this trend was not statistically significant (*p* = 0.096; Figure 2E). Actually, this lack of association could not be exactly explained; there might be several possible reasons. First, the lack of correlation with PD-L1 expression in tumor tissues may be associated with the finding that soluble and exosomal PD-L1 can be generated by various cells, including tumor and blood cells. Second, it is still not clearly determined how the extent of PD-L1 expression in tumor tissue could be assessed because the discrimination of tumor cells with macrophages is still difficult. Accordingly, there are variable cut-off values for the positivity of PD-L1. Third, the number of patients was relatively small to draw a significant association between them. Nevertheless, this lack of association remained as the limitation of our study, and further research should be warranted for better understanding of soluble and exosomal PD-L1 as a predictor of treatment outcomes.

When patients were dichotomized into high and low soluble and exosomal PD-L1 groups, patients with high exosomal PD-L1 were more likely to have refractory disease and a poor performance status (Table 1). These results imply the measurement of soluble and exosomal PD-L1 may influence the outcome after pembrolizumab treatment. However, pre-treatment levels of exosomal and soluble PD-L1 were lower in patients with CR and PR compared with patients with PD, and the response to pembrolizumab was significantly related to pre-treatment low soluble and exosomal PD-L1 (Figure 3A,B). As responders to pembrolizumab maintained their treatments without disease progression, patients with low soluble and exosomal PD-L1 had longer times to progression after treatment with pembrolizumab, leading to longer overall survival than patients with high soluble and exosomal PD-L1 (Figure 3C–E). In the longitudinal assessment during treatment, maintaining a CR was associated with a lower level of soluble and exosomal PD-L1, whereas the opposite case showed increased levels inconsistent with disease progression (Figure 4A,B). This increase in exosomal PD-L1 during PD1-inhibitor treatment may be an adaptive response of tumor cells to T-cell regeneration through interferon-gamma production by reinvigorated CD8+ T cells [32]. Flow cytometry analysis also revealed an increase in CD4 and CD8 T cells in patients who were responders to pembrolizumab compared with non-responders (Figure 5A–C). These results were consistent with the finding that exosomes expressing PD-L1 inhibit T-cell activation in a co-culture of T cells with exosomes, reducing the infiltration of CD4+ and CD8+ T cells into tumor sites [13,33]. As our study found a strong correlation between soluble and exosomal PD-L1, patients with high levels of exosomal PD-L1 may also have high levels of soluble PD-L1. These results suggest exosomal PD-L1 may be a potential therapeutic target, and a recent study reported that the inhibition of exosomal PD-L1 can enhance the anti-tumor immune response with immune checkpoint inhibitors in an animal model [34]. As soluble PD-L1 can act as membrane-bound PD-L1 binding to PD1 in T cells, high levels of circulating soluble PD-L1 could competitively inhibit the effect of PD1 inhibitors such as pembrolizumab (Figure 5D).

## 5. Conclusions

Our study demonstrated that measurements of soluble and exosomal PD-L1 can predict responses to pembrolizumab treatment in patients with ENKTL, and there was a significant association between exosomal and soluble PD-L1. Given that measurements of soluble PD-L1 are more feasible and convenient compared with exosomal PD-L1, pre-treatment soluble PD-L1 may be a useful biomarker for predicting responses to pembrolizumab. Further studies with a larger number of ENKTL patients are warranted to confirm these results.

## Figures and Tables

**Figure 1 cancers-14-05618-f001:**
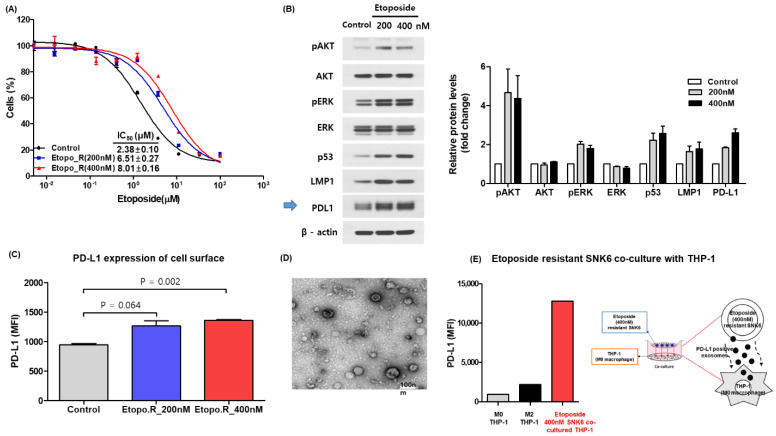
(**A**) Development of etoposide-resistant SNK6R cells. (**B**) Increased expression of PD-L1 in etoposide-resistant SNK6R cells. The original blots could be found in Appendix A. (**C**) Increased expression of PD-L1 on the surfaces of SNK6R cells. (**D**) Transmission electron microscopy identifies exosomes. (**E**) Co-culturing of SNK6R cells with a macrophage cell line (THP-1) reveals an increase in PD-L1.

**Figure 2 cancers-14-05618-f002:**
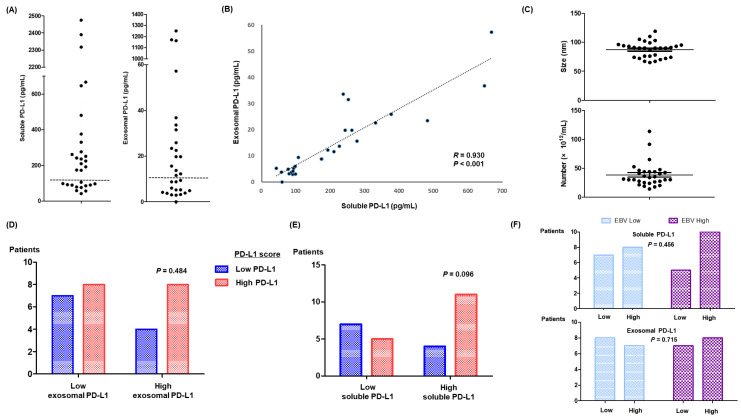
(**A**) Serum levels of soluble and exosomal PD-L1. (**B**) Correlation of exosomal and soluble PD-L1. (**C**) Distribution of the size and number of exosomes. (**D**,**E**) Comparison of PD-L1 scores in tumor tissue according to high and low exosomal and soluble PD-L1 groups. (**F**) The lack of correlation between high titers of EBV DNA and soluble and exosomal PD-L1.

**Figure 3 cancers-14-05618-f003:**
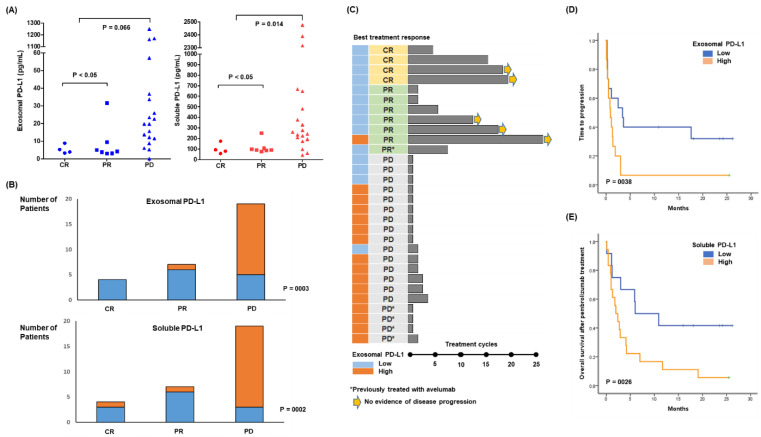
(**A**) Exosomal (blue dots) and soluble (red dots) PD-L1 levels in patients with CR, PR, and PD. (**B**) Comparison of responses based on the high (orange bars) and low (blue bars) exosomal and soluble PD-L1 groups. (**C**) A swimmer plot of 30 patients. (**D**) Comparison of time to progression according to exosomal PD-L1. (**E**) Comparison of overall survival after pembrolizumab treatment between low and high soluble PD-L1 groups.

**Figure 4 cancers-14-05618-f004:**
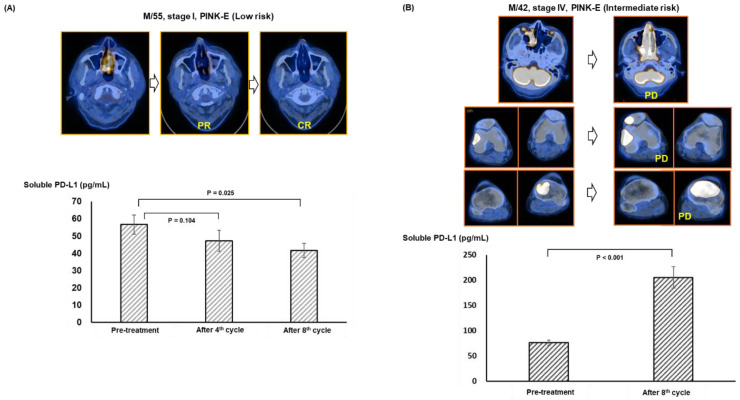
(**A**) A 55-year-old male with stage I and at low risk of PINK-E had a localized ^18F^-FDG uptake lesion in his nasal cavity on PET/CT, and he achieved complete response after the eighth cycle of pembrolizumab treatment (top). His pre-treatment soluble PD-L1 level was significantly lower than that of the complete response (bottom). (**B**) A 42-year-old male with stage IV and at intermediate risk of PINK-E had multiple ^18F^-FDG uptake lesions including both the nasal cavity and the knees on PET/CT, and he failed to respond to pembrolizumab treatment (top). At the time of disease progression after the eighth cycle, his soluble PD-L1 significantly increased compared to that of pre-treatment soluble PD-L1 level (bottom).

**Figure 5 cancers-14-05618-f005:**
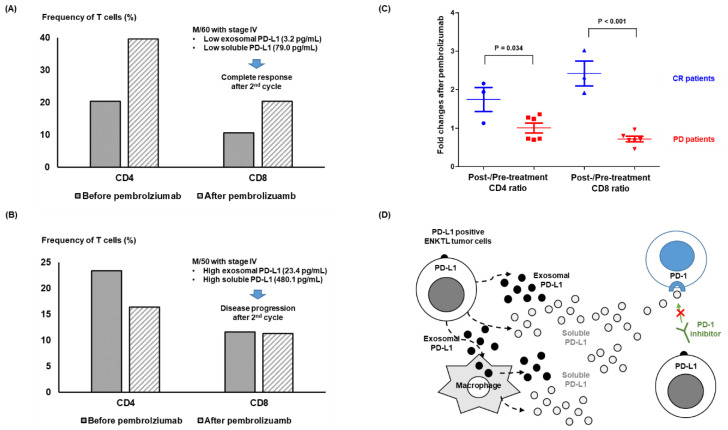
(**A**) A 60-year-old male with low exosomal and soluble PD-L1 achieved a complete response after the second cycle of pembrolizumab. The proportion of CD4 and CD8 cells was compared before and after pembrolizumab in this patient. The flow cytometry showed an increase in CD4 and CD8 T-cells after pembrolizumab treatment. (**B**) A 50-year-old male with high exosomal and soluble PD-L1 showed disease progression after the second cycle of pembrolizumab. The proportion of CD4 cells after pembrolizumab treatment was lower than that of pre-treatment CD4 cells, and the increase of CD8 cells did not occur in this patient, who failed to respond to pembrolizumab. (**C**) Fold changes after pembrolizumab treatment were compared between CR and PD patients. The post-/pre-treatment CD4 and CD8 ratio was significantly higher in CR patients than in PD patients, implying the correlation of response to pembrolizumab with T-cell activation after pembrolizumab treatment. CR: complete response; PD: progressive disease. (**D**) A proposed model of soluble and exosomal PD-L1 influencing the efficacy of PD1 inhibitors in patients with ENKTL.

**Table 1 cancers-14-05618-t001:** Characteristics of patients at the time of pembrolizumab treatment.

Characteristics	Total (n = 30)	Exosomal PD-L1		*p*
		Low	High	
Age (years)				
≤60	23 (77)	11 (73)	12 (80)	>0.99
>60	7 (23)	4 (27)	3 (20)	
Sex				
Male	21 (70)	13 (87)	8 (53)	0.109
Female	9 (30)	2 (13)	7 (47)	
Performance status				
ECOG 0 or 1	13 (43)	10 (67)	3 (20)	0.025
ECOG ≥ 2	17 (57)	5 (33)	12 (80)	
Serum LDH				
Normal	10 (33)	6 (40)	4 (27)	0.700
Increased	20 (67)	9 (60)	11 (73)	
Stage				
I/II	5 (17)	4 (27)	1 (7)	0.330
III/IV	25 (83)	11 (73)	14 (93)	
Disease status				
Relapsed	10 (33)	8 (53)	2 (13)	0.050
Refractory	20 (67)	7 (47)	13 (87)	
PINK-E risk				
Low	3 (10)	2 (13)	1 (7)	0.135
Intermediate	3 (10)	3 (20)	0 (0)	
High	24 (80)	10 (67)	14 (93)	
Number of previous treatments				
<2	18 (60)	10 (67)	8 (53)	0.710
≥2	12 (40)	5 (33)	7 (47)	
Previous treatment with avelumab				
Yes	5 (17)	1 (7)	4 (27)	0.330
No	25 (83)	14 (93)	11 (73)	
Autologous SCT				
Done	13 (43)	6 (40)	7 (47)	>0.99
Not done	17 (57)	9 (60)	8 (53)	

ECOG = Eastern Cooperative Oncology Group; LDH = lactate dehydrogenase; PINK-E = Prognostic Index for Natural Killer/Epstein–Barr virus; SCT = Stem cell transplantation.

## Data Availability

All data will become publicly available upon request from the corresponding authors.

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
