# Peer review of "Exosomal and Soluble Programed Death-Ligand 1 (PD-L1) Predicts Responses to Pembrolizumab in Patients with Extranodal NK/T-Cell Lymphoma"

_cancers, 2022, doi:10.3390/cancers14225618_

Round 1

Reviewer 1 Report

In this paper, the authors investigated the association between pre-treatment soluble and exosomal PD-L1 and outcomes in ENKTL patients who received pembrolizumab as a salvage treatment. Their results revealed that serum-soluble PD-L1 levels were significantly correlated with exosomal PD-L1, and were significantly lower in responders to pembrolizumab compared with non-responders. The results suggest a potential role of soluble and exosomal PD-L1 as a useful pre-treatment biomarker for ENKTL patients receiving pembrolizumab.

As mentioned by the authors, several papers have already published on the prognostic role of soluble and/or exosomal PDL1 as prognostic factors in ENKTL. A study by Feng Y et al further identified soluble PD-L1 as a predictor of treatment response in ENKTL patients and their results suggest that the levels of PD-L1 mRNA in PBMCs and soluble PD-L1 in plasma are useful for ENKTL staging and prediction of treatment response. Nevertheless, this paper has merit as it specifically focuses on the clinical role of soluble and exosomal PDL1 for predicting outcomes after pembrolizumab treatment which, according to the authors, has yet to be studied in ENKTL patients.

1. The above reference should be included:

Feng Y, Jing C, Yu X, Cao X, Xu C. Predicting treatment response of patients with extranodal natural killer/T-cell lymphoma based on levels of PD-L1 mRNA and soluble PD-L1. Hematol Oncol. 2020 Oct;38(4):467-477.

2. In the discussion, the authors stated that “These results imply the measurement of soluble and exosomal PD-L1 can independently influence the outcome after pembrolizumab treatment.” However, there is no multivariate or equivalent analysis to clearly demonstrate that soluble and exosomal PD-L1 expression are independent predictors of responses to pembrolizumab in ENKTL patients?

3. page 5, 3.1: … “expression of phosphorylated AKT and ERK as well as p53 and LMP1 was higher in SNK6R cells (Figure 1A, B).” There is no discussion on the relevance of this finding in this study?

4. The authors reported that the co-culture of SNK6R with THP-1 cells showed higher expression of PD-L1 compared with control (M0 and M1 THP-1) cells, supporting the transfer of exosomal PD-L1 from SNK6R to THP-1 cells (Figure 1G). What is the rationale/relevance of demonstrating possible transfer of exosomal PD-L1 from SNK6R to THP-1 cells? Detection of a higher expression of PD-L1 in this co-culture does not necessarily indicate that there is transfer of exosomal PD-L1 from SNK6R to THP-1 cells.

5. Under flow cytometry, CD56 was performed but nothing is mentioned about CD56 in the manuscript.  

6. Supplementary table 1 included FOXP3: why was this marker included and not reported in the manuscript?

6. The authors used score > 10 to designate high PD-L1 expression based on their previous study on immune subtyping of ENKTL. The basis and derivation of this cut off criteria should be better explained. This is likely an arbitrary cut off and may explain why there is no significant correlation between high levels of soluble PD-L1 (> 100 pg/mL) had a high PD-L1 score. Have the authors tried other cut-off levels to define high PD-L1?

7. The figures, especially figure 1, comprise multiple composite smaller figures which are very small and the font of the axis labels are too small and difficult to read. Suggest to move some figures to supplementary file. For example, Fig 2F can be shifted to supplementary file. Fig 2D and E can be combined into one figure or go to the supplementary file.

8. Figure 3A has no p-value provided.

9. Much of the discussion is a repetition of the results. Discussion on the relevance of the findings can be improved.

Minor comments:

Typographical error: Page 4, 2.6: The pellets where then resuspended in 100 μL of PBS.

Author Response

In this paper, the authors investigated the association between pre-treatment soluble and exosomal PD-L1 and outcomes in ENKTL patients who received pembrolizumab as a salvage treatment. Their results revealed that serum-soluble PD-L1 levels were significantly correlated with exosomal PD-L1, and were significantly lower in responders to pembrolizumab compared with non-responders. The results suggest a potential role of soluble and exosomal PD-L1 as a useful pre-treatment biomarker for ENKTL patients receiving pembrolizumab.

As mentioned by the authors, several papers have already published on the prognostic role of soluble and/or exosomal PDL1 as prognostic factors in ENKTL. A study by Feng Y et al further identified soluble PD-L1 as a predictor of treatment response in ENKTL patients and their results suggest that the levels of PD-L1 mRNA in PBMCs and soluble PD-L1 in plasma are useful for ENKTL staging and prediction of treatment response. Nevertheless, this paper has merit as it specifically focuses on the clinical role of soluble and exosomal PDL1 for predicting outcomes after pembrolizumab treatment which, according to the authors, has yet to be studied in ENKTL patients.

1. The above reference should be included:

Feng Y, Jing C, Yu X, Cao X, Xu C. Predicting treatment response of patients with extranodal natural killer/T-cell lymphoma based on levels of PD-L1 mRNA and soluble PD-L1. Hematol Oncol. 2020 Oct;38(4):467-477.

--> We inserted it as a reference in the section of introduction (Page 2).

2. In the discussion, the authors stated that “These results imply the measurement of soluble and exosomal PD-L1 can independently influence the outcome after pembrolizumab treatment.” However, there is no multivariate or equivalent analysis to clearly demonstrate that soluble and exosomal PD-L1 expression are independent predictors of responses to pembrolizumab in ENKTL patients?

 --> As the number of patients was relatively small, we could not do multi-variate analysis. Thus, we omitted “independently” and revised the sentence (Page 15).

3. page 5, 3.1: … “expression of phosphorylated AKT and ERK as well as p53 and LMP1 was higher in SNK6R cells (Figure 1A, B).” There is no discussion on the relevance of this finding in this study?

--> We added the part in discussion according to your comments (Page 14).

4. The authors reported that the co-culture of SNK6R with THP-1 cells showed higher expression of PD-L1 compared with control (M0 and M1 THP-1) cells, supporting the transfer of exosomal PD-L1 from SNK6R to THP-1 cells (Figure 1G). What is the rationale/relevance of demonstrating possible transfer of exosomal PD-L1 from SNK6R to THP-1 cells? Detection of a higher expression of PD-L1 in this co-culture does not necessarily indicate that there is transfer of exosomal PD-L1 from SNK6R to THP-1 cells.

--> We agree with your opinion, and there are limitations in our results. So, we added the part for our limitation in discussion (Page 15).

5. Under flow cytometry, CD56 was performed but nothing is mentioned about CD56 in the manuscript.  

--> While we were writing the draft, we made a mistake. So, it was our error, and we deleted it from the methods (Page 9). 

6. Supplementary table 1 included FOXP3: why was this marker included and not reported in the manuscript?

 --> That was our error. Actually, FOXP3 staining was used for immune subtyping of ENKTL as we previously reported (Cho, J.; Kim, S.J.; Park, W.Y.; Kim, J.; Woo, J.; Kim, G.; Yoon, S.E.; Ko, Y.H.; Kim, W.S. Immune subtyping of extranodal NK/T-cell lymphoma: a new biomarker and an immune shift during disease progression. Modern pathology: an official journal of the United States and Canadian Academy of Pathology, Inc 2020, 33, 603-615.). However, we do not need to show the results of FOXP3 in this paper, so we omitted it from the supplementary table 1.

7. The authors used score > 10 to designate high PD-L1 expression based on their previous study on immune subtyping of ENKTL. The basis and derivation of this cut off criteria should be better explained. This is likely an arbitrary cut off and may explain why there is no significant correlation between high levels of soluble PD-L1 (> 100 pg/mL) had a high PD-L1 score. Have the authors tried other cut-off levels to define high PD-L1?

--> We tried other cut-off, but results were similar. That was the limitation of our study. We mentioned this part in discussion (Page 16).

8. The figures, especially figure 1, comprise multiple composite smaller figures which are very small and the font of the axis labels are too small and difficult to read. Suggest to move some figures to supplementary file. For example, Fig 2F can be shifted to supplementary file. Fig 2D and E can be combined into one figure or go to the supplementary file.

--> We moved figures into supplementary figures according to your comments.

9. Figure 3A has no p-value provided.

--> We added p-value according to your comments.

10. Much of the discussion is a repetition of the results. Discussion on the relevance of the findings can be improved.

 --> We revised the discussion according to your comments.

Minor comments:

Typographical error: Page 4, 2.6: The pellets where then resuspended in 100 μL of PBS.

--> We revised it.

Reviewer 2 Report

The manuscript addresses the potential of soluble and exosomal PD-L1 as prognostic marker for the response to the treatment with pembrolizumab in NK/T-cell lymphoma. Actually, this had been previously studied, albeit in patients treated with etoposide, ifosfamide, cisplatin and dexamethasone (Li et al., Am J Cancer Res, 2020). From this point of view, the manuscript has a high degree of novelty. The manuscript consists of a first (in vitro) part, where the authors demonstrate an increase in soluble and PD-L1 expression in cells exposed to etoposide. Subsequently, the authors investigate the association between the levels of soluble and exosomal PD-L1 in serum before the treatment and the treatment response. The experimental design is appropriate. However, the data obtained does not fully support a strong association between PD-L1 and the response to treatment. The following aspects need to be addressed before further consideration of the manuscript: 

1-     A critical aspect is the lack of statistical analysis for most of the data presented in the manuscript. Namely: Figure 1 (all the panels lack statistical analysis), Figure 2 (it is not clear which correlation analysis was performed), Figure 3 (panel A), Figure 4 (all panels) and Figure 5 (all panels). The authors must perform appropriate statistical analysis depending on the number of experimental groups involved. 

2-     The authors must critically discuss the data in terms of the lack of significant association between soluble PD-L1 and PD-L1 in the tumor tissue (section 3.2.).  A lack of association, as presented in the manuscript, argues against the potential of soluble PD-L1 as marker and as relevant mechanistic factor involved in the response to the treatment. 

3-     Section 2.5. Western blot. Please, indicate the dilutions of all the antibodies used. 

4-     Section 2.6. This section describes the isolation of exosomes from serum. The procedure for exosome isolation from cell culture medium also needs to be described. 

5-     Section 2.8. Please, indicate the dilutions of the antibodies used for flow cytometry. 

6-     In general, following the requirements of the ISEV, analysis of isolated exosomes with a single-particle method (e.g., NTA, TEM) should be provided. 

7-     Figure 1B. A quantitative analysis of western blot data with at least three biological replicates has be performed.  

8-     Figure 1A. Please, specify the cell type for each IC50 value provided. 

9-     Figure 1G. The coculture experiment requires further clarification. Please, specify the characteristics of the different populations (M0, M1, M2).

10-  Section 3.2. PD-L1 data in healthy volunteers need to be shown. 

11-  Section 3.3. Data on the association between soluble PD-L1 and time to progression and pretreatment exosomal PD-L1 and post-treatment overall survival, currently not shown, need to be included in the manuscript.

12-  Figure 4. Images presented at the bottom of each panel need to be described in the figure legend. 

Author Response

The manuscript addresses the potential of soluble and exosomal PD-L1 as prognostic marker for the response to the treatment with pembrolizumab in NK/T-cell lymphoma. Actually, this had been previously studied, albeit in patients treated with etoposide, ifosfamide, cisplatin and dexamethasone (Li et al., Am J Cancer Res, 2020). From this point of view, the manuscript has a high degree of novelty. The manuscript consists of a first (in vitro) part, where the authors demonstrate an increase in soluble and PD-L1 expression in cells exposed to etoposide. Subsequently, the authors investigate the association between the levels of soluble and exosomal PD-L1 in serum before the treatment and the treatment response. The experimental design is appropriate. However, the data obtained does not fully support a strong association between PD-L1 and the response to treatment. The following aspects need to be addressed before further consideration of the manuscript: 

1. A critical aspect is the lack of statistical analysis for most of the data presented in the manuscript. Namely: Figure 1 (all the panels lack statistical analysis), Figure 2 (it is not clear which correlation analysis was performed), Figure 3 (panel A), Figure 4 (all panels) and Figure 5 (all panels). The authors must perform appropriate statistical analysis depending on the number of experimental groups involved. 

--> We added the p-value on figure 1C, and we showed the quantitative analysis of western blot data in figure 1B and D. In figure 2, we did Spearman correlation analysis, and we added it in the methods. We added p value according to your comments in figure 3A. In figure 4, we added p value according to your comments, and revised the figure legends. As the pattern of soluble and exosomal PD-L1 was similar, we showed the changes of soluble PD-L1. In figure 5, we presented the representative case of CR patient in figure 5A. In this figure, we showed increase of CD4 and CD8 proportion after pembrolizumab treatment suggesting the correlation between T-cell activation and response to pembrolizumab. In figure B, the increase of CD4 and CD8 were not observed in a patient who failed to respond to pembrolizumab. As these two figures represented each case, so we could not add p-value. However, we added p value after we did unpaired t-test between CR and PR patients in figure 5C according to your comments. Thus, we revised the figure legends for figure 5.   

2. The authors must critically discuss the data in terms of the lack of significant association between soluble PD-L1 and PD-L1 in the tumor tissue (section 3.2.).  A lack of association, as presented in the manuscript, argues against the potential of soluble PD-L1 as marker and as relevant mechanistic factor involved in the response to the treatment. 

--> We agree with your opinion, so we added the limited value of our study in discussion (Page 16).

3.  Section 2.5. Western blot. Please, indicate the dilutions of all the antibodies used. 

--> We revised the methods according to your comments (Page 7).

4. Section 2.6. This section describes the isolation of exosomes from serum. The procedure for exosome isolation from cell culture medium also needs to be described. 

--> We revised the methods according to your comments (Page 8).

5. Section 2.8. Please, indicate the dilutions of the antibodies used for flow cytometry. 

--> For the description of flow cytometry, we usually described the volume of each antibody that were used for analyses. In this experiment, we used 5 uL of each antibody for reaction. Thus, we revised the sentence according to your comments (Page 9).

6. In general, following the requirements of the ISEV, analysis of isolated exosomes with a single-particle method (e.g., NTA, TEM) should be provided. 

--> We added the picture of TEM in figure 1 and added the methods for NTA and TEM (Page 9).

7. Figure 1B. A quantitative analysis of western blot data with at least three biological replicates has be performed.  

--> We revised the figure 1B according to your comments.

8. Figure 1A. Please, specify the cell type for each IC50 value provided. 

--> We revised the figure 1A according to your comments.

9. Figure 1G. The coculture experiment requires further clarification. Please, specify the characteristics of the different populations (M0, M1, M2).

--> We revised the methods according to your comments (Page 6).

10. Section 3.2. PD-L1 data in healthy volunteers need to be shown. 

--> There was no PD-L1 data of healthy volunteers because we could not obtain tissue samples from them. Thus, we only presented their soluble and exosomal PD-L1 data as a control group (Page 12).

11. Section 3.3. Data on the association between soluble PD-L1 and time to progression and pretreatment exosomal PD-L1 and post-treatment overall survival, currently not shown, need to be included in the manuscript.

--> We added the data and revised the sentences as you recommended (Page 13).

12. Figure 4. Images presented at the bottom of each panel need to be described in the figure legend. 

--> We revised the figure legends as you recommended.  

Round 2

Reviewer 2 Report

The authors have addressed all my concerns.